# Implicit Solutions of the Electrical Impedance Tomography Inverse Problem in the Continuous Domain with Deep Neural Networks

**DOI:** 10.3390/e25030493

**Published:** 2023-03-13

**Authors:** Thilo Strauss, Taufiquar Khan

**Affiliations:** 1Research Department at ETAS GmbH, Robert Bosch GmbH, 70469 Stuttgart, Germany; 2Department of Mathematics and Statistics, University of North Carolina at Charlotte, Charlotte, NC 28223, USA

**Keywords:** inverse problems, electrical impedance tomography, implicit solutions, deep learning, neural networks, continuous domains

## Abstract

Electrical impedance tomography (EIT) is a non-invasive imaging modality used for estimating the conductivity of an object Ω from boundary electrode measurements. In recent years, researchers achieved substantial progress in analytical and numerical methods for the EIT inverse problem. Despite the success, numerical instability is still a major hurdle due to many factors, including the discretization error of the problem. Furthermore, most algorithms with good performance are relatively time consuming and do not allow real-time applications. In our approach, the goal is to separate the unknown conductivity into two regions, namely the region of homogeneous background conductivity and the region of non-homogeneous conductivity. Therefore, we pose and solve the problem of shape reconstruction using machine learning. We propose a novel and simple jet intriguing neural network architecture capable of solving the EIT inverse problem. It addresses previous difficulties, including instability, and is easily adaptable to other ill-posed coefficient inverse problems. That is, the proposed model estimates the probability for a point of whether the conductivity belongs to the background region or to the non-homogeneous region on the continuous space $R^{d}\cap \Omega$ with d∈{2,3}. The proposed model does not make assumptions about the forward model and allows for solving the inverse problem in real time. The proposed machine learning approach for shape reconstruction is also used to improve gradient-based methods for estimating the unknown conductivity. In this paper, we propose a piece-wise constant reconstruction method that is novel in the inverse problem setting but inspired by recent approaches from the 3D vision community. We also extend this method into a novel constrained reconstruction method. We present extensive numerical experiments to show the performance of the architecture and compare the proposed method with previous analytic algorithms, mainly the monotonicity-based shape reconstruction algorithm and iteratively regularized Gauss–Newton method.

## 1. Introduction

The electrical impedance tomography (EIT) inverse problem is a severely ill-posed inverse problem that attempts to reconstruct the unknown conductivity of an object Ω from boundary electrode measurements. EIT is a great alternative to CT scans, particularly for neonatal brain imaging, where CT can be harmful. EIT has been successfully used in medical imaging [1,2], monitoring soil [3], or crack detection [4]. The EIT forward problem is an elliptic differential equation:(1)$$-\nabla \cdot \left(\sigma \nabla u\right)=0\;\mathrm{in}\;\Omega ,$$
where $u\in H^{1}\left(\Omega \right)$ is the electric potential, $\sigma \in L^{\infty }\left(\Omega \right)$ is the known conductivity with $0<\sigma_{1}\leq \sigma \leq \sigma_{2}<\infty$, and there are no current sources inside Ω. We assume the electric conductivity σ is a bounded, positive, and isotropic conductivity distribution with $\sigma \geq \sigma_{0}>0$ for some $\sigma_{0}\in R$. Before solving (Equation 1), we must formulate a forward model that incorporates boundary conditions about the EIT forward problem. Once we have the EIT forward model, we can proceed to solve its inverse problem. In theory, the EIT inverse problem is exponentially ill posed and highly nonlinear. Even if a solution exists for (Equation 1), it will be highly sensitive to any noise or perturbations in the data. Therefore, in practice, we must impose a priori knowledge through some form of regularization. In the past, research topics in EIT mainly focused on analytical and numerical methods. Typically, the conductivity is discretized into triangles (2D) or tetrahedrons (3D). This discretization allows solving the PDE of the forward problem via the finite element method (FEM).

To address the poorly posed EIT inverse problem, significant research has been dedicated to developing deterministic and statistical reconstruction methods to improve the recovered tomographic images [5,6,7,8,9]. Deterministic and statistical methods enforce regularization to provide reasonable image reconstructions. However, in many applications, tomographic images of decent quality do not suffice. Often, high-resolution tomographic reconstructions are required for accurate diagnostic analyses or the detection of anomalies.

### 1.1. Contributions

We propose a new formulation of the EIT inverse problem. In particular, instead of reconstructing the conductivity function at each point, we find a function that predicts if a point p∈Ω belongs to the background conductivity or not. Here, the background conductivity refers to the known conductivity that the homogeneous portion of Ω should have. The assumption that such conductivity is known is also common in the analytical/numerical setting. Some regularization use this conductivity as an a priori assumption for parametrization [7]. Furthermore, most optimization algorithms require some knowledge of the background conductivity as an initialization of the unknown conductivity [10].

Our goal is to define a function $f:R^{d}\to \left\{0,1\right\}$ with d∈{2,3} that for any point $p\in R^{d}$ estimates if *p* is a background conductivity. We parametrize *f* with a neural network. In detail, this neural network requires three network blocks. We call the first neural network the measure encoder $f_{measure}$. It takes the measurements *m* of an EIT experiment and maps them to a measurement embedding $m_{e}\in M\subseteq R^{dim(m_{e})}$. Here, $dim(m_{e})$ refers to the dimension of the embedding space. A second neural network called the point encoder $f_{point}$ takes a point $p\in R^{d}$ and maps it to a point embedding $p_{e}$. We call the third network the decoder network $f_{decoder}$. It takes the point embedding conditioned on measure embedding to predict if a point *p* is from the background conductivity. In summary, we write the novel network architecture as $f:R^{d}\times M\to \left\{0,1\right\}$. See Figure 1 for a visualization of the method. Observe that for solving one inverse problem, the measurement embedding $m_{e}$ only has to be computed once. On the other hand, we compute the point encoder $f_{point}$ and the decoder network $f_{decoder}$ for every point $p\in R^{d}$ on that for which we solve the inverse problem.

To show the use of the shape estimation method for reconstructing the unknown conductivity distribution σ, we combine it with gradient-based optimization methods. In detail, we introduce a constraint and a piecewise optimization problem.

In summary, we make the following contributions:We present a novel method to solve the EIT shape estimation problem in a continuous domain using machine learning.We introduce a constraint and a piece-wise gradient-based optimization problem by using the shape reconstruction from the proposed machine learning algorithm.We show that the proposed method using machine learning outperforms the benchmark approaches which do not involve machine learning.

### 1.2. Related Work

An EIT experiment involves applying an electrical current (Neumann data) on ∂Ω, the boundary of the body Ω, to measure the corresponding electrical potential differences on ∂Ω. Hereby, the information on the Neumann-to-Dirichlet (NtD) operator $\Lambda_{\sigma }$ is obtained. The estimate of the unknown conductivity σ is then reconstructed from a set of EIT experiments [11,12,13,14].

The are many applications of EIT, including medical imaging [1,2], monitoring soil [3], or crack detection [4]. There is also work on hardware design in related problems [15]. It is well known that EIT is highly nonlinear and strongly ill posed. Therefore, EIT requires regularization for proper reconstruction [5,6]. In the literature, there are abundant deterministic approaches for solving the EIT inverse problem, including the factorization method [16], d-bar method [17], or variational methods for least-square fitting [5,6,10,18]. There also exists previous work on statistical methods for solving the EIT problem [7,10,19,20]. There are deterministic approaches to reconstructing the shape of the abnormalities, for example, the monotonicity shape estimate [21] or the level set method [22,23]. Due to the simplicity and computational speed, we choose to compare our method in this paper with the monotonicity shape estimate [21] and iteratively regularized Gauss–Newton (IRGN) method [10].

With the rise of deep learning, it has become state of the art in many tasks, such as image recognition [24], strategy board games [25], or protein folding problems [26]. While these results have been impressive, deep learning did not become state of the art in EIT or other ill-posed inverse problems. However, solving the EIT inverse problem with deep learning is receiving increasing attention, for example, with radial bases functions [27], fully connected neural networks [28], and in particular by using convolutional neural networks [29,30,31]. While convolutional neural networks are attractive in the 2D inverse problem, they start having a limitation in resolution when solving the 3D problem. This limitation is because the number of voxels grows cubically as does the computational cost of the voxel representation. In 3D computer vision, many applications work in a voxelization of usually around $32^{3}$ or $64^{3}$ [32]. It is questionable whether this is sufficient for applications of EIT. Since 2019, in the computer vision community, implicit representations of 3D objects via neural networks replaced voxel-based and mesh-based methods as the state of the art for several applications. Some of these results include single image 3D reconstructions [33,34], representing texture on 3D objects [32], representing surface light fields in 3D [35], or 3D reconstructions from many images [36].

### 1.3. Rationale for Proposed Shape Reconstruction Using Machine Learning

It is well known that the EIT shape reconstruction inverse problem for piecewise-constant conductivity suffers from logarithmic instability as shown in [37]. The stability of the shape reconstruction for the EIT inverse problem also depends on the type of assumptions on the parameter space and the analytical technique used, for example, if we consider conductivity that belongs to a finite-dimensional set of piecewise-analytic functions that are bounded from above and below by a priori known constants and use theoretical technique using monotonocity. Then the stability is of the type Lipschitz [21]. There is no constructive estimate on *C*, and the assumption is that infinite information on the Dirichlet-to-Neumann map Λ is known. Here, *C* refers to the right hand side constant used in the following Lipschitz stability inequality $\left|\right|\sigma_{1}-\sigma_{2}{\left|\right|}_{L^{\infty }\left(\Omega \right)}\leq C||\Lambda_{\sigma_{1}}-\Lambda_{\sigma_{2}}{\left|\right|}_{*}$, where ∗ represents the operator norm of $\Lambda :L_{⋄}^{2}\left(\partial \Omega \right)\to L_{⋄}^{2}\left(\partial \Omega \right)$ and $L_{⋄}^{2}\left(\partial \Omega \right)$ is the $L^{2}\left(\partial \Omega \right)$ space with vanishing integral mean on the boundary [38].

Using a completely different approach using the Bayesian formulation, one can obtain the (locally) Lipschitz stability of the probability distribution of unknown domains for the general shape identification inverse problem for a heat cavity problem that in principle should also be applicable to EIT [39]. Therefore, the proposed machine learning approach combining both statistical and deterministic shape reconstruction techniques should provide better stability. The focus of this manuscript is to demonstrate the efficacy of the proposed approach computationally.

## 2. Materials and Methods

In this section, we first describe the preliminary neural network blocks. Hereby, we aim to unify the basic knowledge of the terms needed to describe our network architecture. Then we describe the proposed network architecture in detail. This is followed by an explanation of how the training data are generated, what noise setting is used, and how the neural network is trained.

### 2.1. Leaky Rectified Linear Units

Leaky rectified linear units (Leaky ReLu) [40] is a nonlinear function used as an activation function in our architecture. We write Leaky ReLu as
$$\mathrm{LeakyReLu}\left(x\right):=\left\{\begin{array}{cc} 0.01x & \mathrm{if}\;x<0\\ x & \mathrm{else}. \end{array}\right.$$

### 2.2. Residual Neural Network Blocks

A residual neural network (ResNet) is a neural network with shortcuts between layers [41]. We use fully connected ResNet blocks with Leaky ReLu activations. It is among the most common building block in state-of-the-art networks. Here, we use the following variant: if the input and output sizes differ, we use a single fully connected network as a shortcut instead of copying the input data. Find a visualization of ResNet in Figure 2.

### 2.3. Softmax

One uses the softmax function to convert an array $x=(x_{1},...,x_{n})$ into a probability distribution. We write softmax as
$$\mathrm{softmax}\left(x\right):=\left(\frac{exp(x_{1})}{\sum_{i}exp\left(x_{i}\right)},...,\frac{exp(x_{n})}{\sum_{i}exp\left(x_{i}\right)}\right).$$

### 2.4. Implicit Network Architecture for EIT

The proposed network architecture requires three network blocks that operate on each other. Find the visualization of the network architecture in Figure 1.

When we perform an EIT experiment, we obtain the boundary electrode measurements *m*. We represent these measurements in the form of an array. Then, we use a network architecture called the measure encoder network $f_{measure}$ to map *m* to its corresponding embedding $m_{e}\in M\subseteq R^{dim(m_{e})}$. Here, the embedding $m_{e}$ is an array that stores the relevant information of the measurements. We write the $f_{measure}$ as a series of ResNet blocks. See Table 1 for the exact setup.

For us, to solve the EIT inverse problem implicitly means to estimate if a point $p\in R^{d}\cap \Omega$ with d∈{2,3} belongs to the background conductivity. To find this estimate, we first define a point encoder $f_{point}$. The point encoder takes a point $p\in R^{d}\cap \Omega$ and maps it to a point embedding $p_{e}$. The point embedding is an array of the same dimensionality as the measure embedding $m_{e}$. We describe the architecture of the point encoder $f_{point}$ in Table 2.

We call the last part of the implicit architecture the decoder network $f_{decoder}$. The input array of $f_{decoder}$ is computed by adding the point and measure embedding $p_{e}+m_{e}$. Hence, we condition the point embedding $p_{e}$ on the measure embedding $m_{e}$. With the conditioning term, we want to emphasize that the measure embedding only needs to be computed once, while we compute the point embedding for every point at which we solve the inverse problem. The decoder network consists of a series of ResNet blocks mapping to a two-dimensional array. We apply the softmax function to the output to obtain a probability. Then the network predicts that the point is in the background if the first probability is larger than the second in the array. Otherwise, it predicts that the point is anomalous. The network architecture is summarized in Table 3.

### 2.5. Generating Training Data

In this section, we briefly describe the well-studied EIT forward problem. We emphasize that, here, our goal is not to provide a complete description of the forward problem. Refer to [11,12,13] for a theoretical discussion and [42] for a description of the the finite element method. Based on the EIT forward problem, we describe how to generate data suitable for training the network architecture in Figure 1.

### 2.6. The EIT Forward Problem

The EIT forward problem finds the boundary electrode measurements for a known conductivity distribution σ on Ω. We write the partial differential equation as
$$\begin{array}{cc} \; & \nabla \cdot (\sigma \nabla u)=0\;\mathrm{on}\;\Omega \\ \; & \sigma \frac{\partial u}{\partial n}=g_{1}\;\mathrm{on}\;\partial \Omega \;\;(\mathrm{Neumann}\;\mathrm{Input}\;\mathrm{Current})\\ \; & u=g_{2}\;\mathrm{on}\;\partial \Omega \;\;(\mathrm{Dirichlet}\;\mathrm{Measured}\;\mathrm{Voltage})\\ \; & \int_{\partial \Omega }u=0. \end{array}$$
where $g_{1}$ is the injected current and $g_{2}$ is the measured voltage.

Our experiments use a simple electrode model, where current flows into Ω through a set of boundary electrodes. Specifically, we use a point electrode model, where we use one electrode as a source and its right-hand neighbor as a sink and repeat this for each possible setup. For one source-sink setting, we compute the voltage difference between each electrode and its right-hand neighbor if none of the electrodes is a source nor a sink electrode. Our experiments used 16 electrodes on ∂Ω. Hence, we have 13 measurements per source–sink setting, which results in 208 voltage differences for all source–sink settings that we call the measurements *m*.

As described in [42], the finite element method solves the EIT forward problem for a known into a mesh discretized conductivity distribution σ. However, to train the proposed network architecture, one needs the measurements *m*, a point $p\in R^{d}\cap \Omega$, and whether or not the *p* belongs to the background conductivity.

### 2.7. Sampling Point Clouds

To train our neural network, we need to generate a data set of points and their labels. One can sample a point cloud from $R^{d}\cap \Omega$ in many different ways, for example, by taking uniform distributed points. We found that uniform distributed points do not produce the best performance for training the network. For abnormalities with a small area, few elements of the point cloud will be within that anomaly. Hence, the resulting network is not strong in detecting these abnormalities.

The object Ω is separated into the background and abnormal strata. We sample uniformly the same number of points from each group. In the experimental part, we used 512 points per group of a given conductivity σ. This sampling procedure lets the neural network better perform for detecting small abnormal areas correctly.

### 2.8. Data Sets

We generate a training dataset consisting of 16,384 discretized conductivities and their corresponding measurements ${\left\{m_{i}\right\}}_{i=1}^{16,384}$. The training data set contains 4096 conductivities with one, two, three, and four circular abnormalities, respectively. We randomly chose the radius and the conductivities of each circular object. The background conductivities are constant 1 in the entire dataset. Similarly, we build an analog validation data set of 1024 conductivities and their measurements *m*. Finally, we handcraft a small test data set to compare our methods with baseline approaches.

### 2.9. Noise

In this document, Gaussian measurement noise of a specific noise level of δ% refers to
$$m_{noisy}=m+N_{Size(m)}\left(0,\max\left(m\right)\right)*\frac{\delta }{100}.$$
Note that *m* refers to the measurement array of length 208.

### 2.10. Training

In this section, we describe the training procedure of the proposed neural network architecture. We use the cross-entropy loss [43],
loss(y,q)=−(ylog(q)+(1−y)log(1−q)),
where *y* is the binary indicator of a point being anomalous and *q* the computed probability of being anomalous. All neural networks are trained for 1000 epochs. One epoch refers to moving once over the entire training data set. We use batch size 64. During training for each batch element, we sample a point cloud of 1024 points. Hence, one batch requires 64 forward operations of the measure encoder $f_{measure}$, while it requires 65,536 forward operations of the point encoder $f_{point}$ and the decoder $f_{decoder}$.

All measurements *m* are 0–1 normalized with the maximum–minimum from the training data. We also 0–1 normalize the points *p*. During training, different results, some with and some without noise, are presented. Training with noise implies that we add new Gaussian noise to each measurement *m* in each epoch.

For training, the ADAM optimizer [44] is used with a starting learning rate of $5e^{-4}$. We reset the optimizer every ten epochs and multiply the learning rate by 2/3. All parameters provided in this section aim to make the method reproducible. Most parameters were chosen ad hoc, but some key parameters, such as the learning rate, were iteratively improved to increase the training speed/stabiliy. We trained all networks on a single Nvidia GeForce RTX 2060.

## 3. Results

In this section, we present our experimental results. We first show some numerical results on the training noise and an ablation study on the architecture choice. Then the network architecture is compared with baseline methods.

### 3.1. Noise Study

In a first experiment, we evaluate what level of training noise lets the network architecture perform best on the validation data set. Hence, we train multiple different models by adding different levels of noise to the training data. The noise is generated independently in every training epoch. For simplicity, we consider the network architecture with a scale of 2. Please see Table 1, Table 2 and Table 3 for the meaning of the scale value.

In Table 4, we compute the accuracy and standard deviation by evaluating the network on the center of the triangles of the original mesh. Hereby, we do not commit an inverse crime because the network is capable of solving the inverse problem in the entire domain $R^{2}\cap \Omega$. The accuracy for one measurement *m* is the ratio of points correctly labeled as either background or inclusion, i.e., accuracy $=\frac{\sum_{i\in points}I\left({\mathrm{Prediction}}_{i}=={\mathrm{Truth}}_{i}\right)}{\mathrm{Number}\;\mathrm{of}\;\mathrm{points}}$, where *I* is the indication function. We observe that the network has the best accuracy with a training noise of around 0.1%. For validation noise levels over 3%, larger training noise increases the performance by reducing the standard deviation. See Figure 3 for several reconstructions of the validation data set.

For the rest of this document, we train all networks with 0.1% training noise. Note that we considered 5% as the largest noise level for the validation and test data because the reconstructions tend to become unstable, both the location and shape of the inclusions, for larger noise levels.

### 3.2. Ablation Study

Here, we provide a comparison of several architecture choices. For simplicity, we only present experiments on different scale values (see Table 1, Table 2 and Table 3) for the measurement noise level of 0.1%. Hence, we trained a network independently on each scale value of 1, 2, and 4.

In Table 5, we observe that the architecture with scale 2 outperforms the other scale values for all validation noise levels. Hence, for all further experiments, we use the scale value 2. See Figure 3 for several reconstructions of the validation data set with scale 2. Observe that the reconstructions are relatively robust. As expected, they become less accurate in positioning and shape of the anomalies for larger noise levels.

### 3.3. Comparison with Baseline Approaches

In this section, we compare our method with several benchmark algorithms on simulated test conductivities. Note that the validation data set contains conductivities generated from the same distribution as the training data set. This does not hold for the handcrafted test conductivities. Hence, in this section, we also test the generalization capabilities of the network.

#### 3.3.1. Comparison with Monotonicity-Based Shape Reconstruction

In this paper, we compare our method with the algorithm proposed in [21]. It uses a monotonicity test that comes from the simple relation that for two conductivities $\sigma_{1}$ and $\sigma_{2}$ we have that if $\sigma_{1}\leq \sigma_{2}$ implies $\Lambda_{\sigma_{1}}\geq \Lambda_{\sigma_{2}}$ where Λ is the Neumann-to-Dirichlet (NtD) operator. The second term is to be understood as $\Lambda_{\sigma_{1}}-\Lambda_{\sigma_{2}}$ being positive semi-definite. The idea is to test if the equation holds for many different small balls. Thus, we tested which balls are inside the anomalous region of the unknown conductivity distribution of interest. See [21] for a detailed description of the theory and algorithm. To speed up the method, we precomputed the NtD operator of all small balls on that we perform the test.

In Figure 4, it can be seen that our proposed method outperforms the monotonicity approach in all examples. The monotonicity-based shape reconstruction approach was only evaluated on 0% measurement noise. This is because even for small noise there is typically no positive semi-definite matrix in the tests, thus predicting that there is no anomalous area anywhere.From the numerical results, the proposed method also outperforms the monotonicity test in almost all examples (see Table 6). The computational speed is about 25 times faster using the deep learning approach. In detail, our approach took about 0.007 s while solving the monotonicity test, which about 0.178 s.

#### 3.3.2. Comparison with Iteratively Regularized Gauss-Newton Method

In this section, we compare with the iteratively regularized Gauss–Newton (IRGN) method as described in [10] with IRGN methods that make use of our shape reconstructions. In detail, we reconstruct the true conductivity on a reconstruction mesh. First, we use the standard Tikhonov-regularized IRGN method, see [10] for a good description of the method. Then, we use a constrained IRGN method. Here, constrained means that we first reconstruct the problem with our shape reconstruction approach. Then, the IRGN method is used to reconstruct the mesh only on the abnormal areas of the predicted shape. Lastly, we use the piecewise IRGN method. By piecewise, we mean that we use the proposed shape reconstruction algorithm and optimize all abnormal areas as separate conductivity parameters, thus obtaining a piecewise constant reconstruction.

Some reconstructions can be seen in Figure 5 and the corresponding numerical results in Table 7. We see that both visually and numerically, it is advantageous to incorporate the shape reconstruction into the IRGN method. The constraint method outperforms the other methods numerically. Visually, the piecewise IRNG method seems to be the most informative.

### 3.4. Super-Resolution Shape Reconstruction

In the previous experiments, we only evaluated our method on the reconstruction mesh for providing a fair comparison. Here we use, similarly to the IRGN experiments in Figure 5, a much finer mesh than the one on that which our method is trained. We then evaluate our method on the same mesh. We note that this is not an inverse crime because our method directly solves the problem in the continuous domain.

In Figure 6, we see that even though the method was trained on data from a less fine mesh. It performs very well on data on the finer mesh. While for large noise levels the shape becomes less accurate, it still finds the correct position of the anomaly. This indicates that one could train the network with easily accessible simulated data and still obtain good performance on real experimental data.

### 3.5. Difficulties of the Proposed Method

The main difficulty of the proposed method is that it assumes that the background conductivity is known. However, we note that many other methods also assume this to some extent. Further, problems with this and any other method based on neural networks are that they require an extensive training set to perform well. This is, in particular, problematic when one wants to use real training data because it is costly and time intensive to create a large number of phantoms and measurements for such an approach. Hence, our training is entirely on simulated training data.

## 4. Discussion and Conclusions

We presented a novel neural network-based approach for reconstructing the shape of anomalies for the EIT inverse problem. The method is capable of solving the problem implicitly in the continuous domain. We discussed the theoretical results on the level of ill posedness of the shape reconstruction algorithm to provide some context for the proposed method. The proposed design flow can be used for any coefficient inverse problem, such as EIT. It can be easily applied to any inverse problem that aims to reconstruct unknown images or physical properties from some measurements. The only requirement for using this method is the existence of a properly defined forward model.

Numerical experiments showed that low measurement noise during training the neural network improved the performance on noisy test data. Further numerical experiments with different network parameters were performed to find a good network architecture. Our shape reconstruction algorithm outperformed the monotonicity-based shape reconstruction numerically, visually, and in terms of computational speed. We also proposed using the reconstructed shapes to improve gradient-based methods. In particular, we presented a piecewise and a constrained iteratively regularized Gauss–Newton (IRGN) method. We found that the constrained method outperforms the piecewise and the standard IRGN algorithm numerically. However, the piecewise method was visually the most informative and provided a piecewise constant reconstruction of the conductivity. Furthermore, we presented high-resolution reconstruction with different noise levels and found that the reconstructions using the proposed algorithm are very robust.

In summary, we demonstrate the efficacy of our approach. The proposed approach is simple, easy to transfer to other inverse problems, efficient in terms of computational speed, and able to solve the EIT problem directly in the continuous domain.

## Figures and Tables

**Figure 1 entropy-25-00493-f001:**
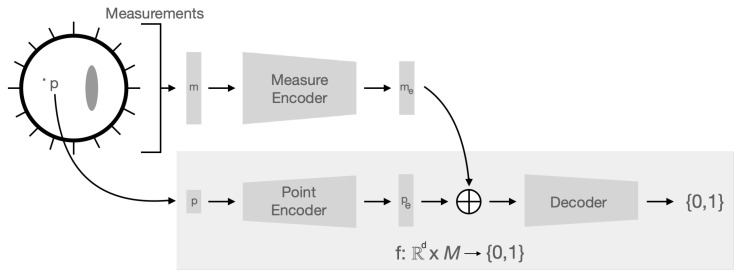
The image represents a schematic description of the proposed network architecture. Here, d∈{2,3} is the dimensionality of the EIT inverse problem. The implicit architecture is based on two main fully connected networks, a measurement encoder tanking measurements as input and a point encoder that has a point in $R^{d}$ as input. The outputs are added. Then a fully connected decoder network predicts the probability of a point (the input of the point encoder) being of the background or anomaly type.

**Figure 2 entropy-25-00493-f002:**
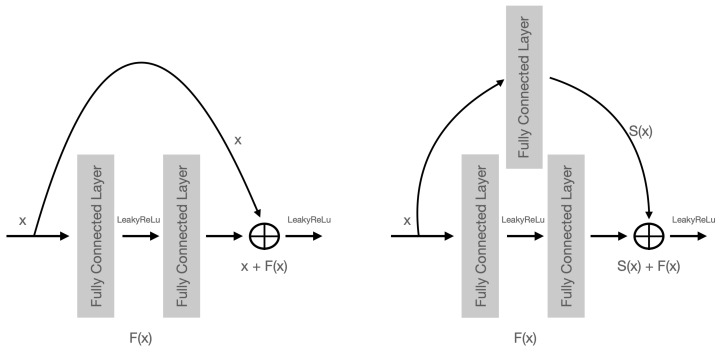
A schematic picture of the residual neural network (ResNet) versions that we use as the main building block in our implicit architecture. On the left-hand side is the standard ResNet block with Leaky ReLu activation. On the right-hand is a modified ResNet block that can handle different input–output sizes.

**Figure 3 entropy-25-00493-f003:**
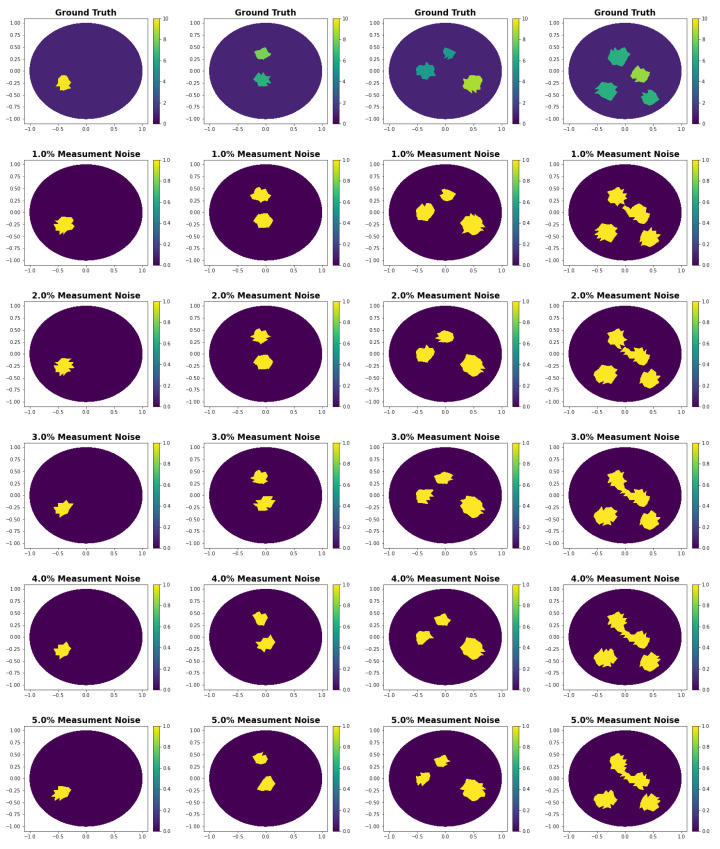
Shape reconstruction using the proposed algorithm with scale 2 and training noise 0.1%. The ground truth images are randomly chosen from the validation data set with 1, 2, 3, and 4 inclusion. The reconstructions are performed at different noise levels. They are evaluated at points at the center of the triangles of the ground truth mesh. Hereby, we do not commit an inverse crime because our method solves the problem on the continuous domain.

**Figure 4 entropy-25-00493-f004:**
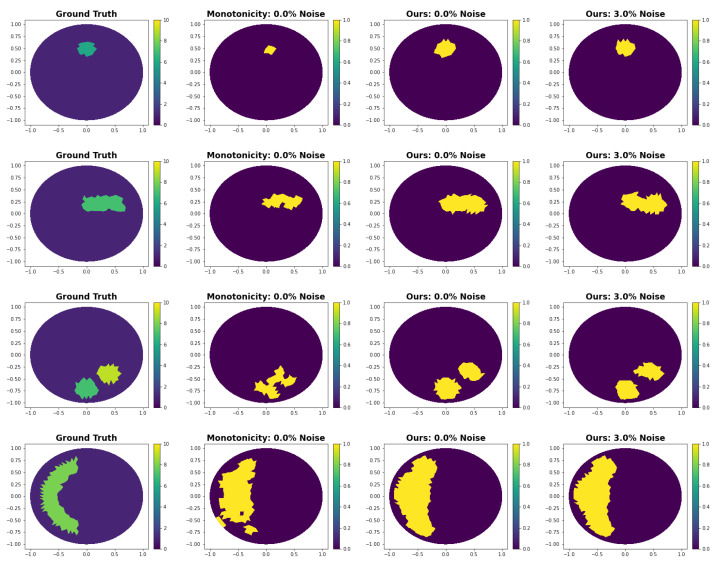
Shape reconstruction comparing our method with the monotonicity-based shape reconstruction algorithm. Due to the noise sensitivity of the monotonicity method, we only evaluated it at a noise level of 0.0% and with the same simulation and reconstruction mesh.

**Figure 5 entropy-25-00493-f005:**
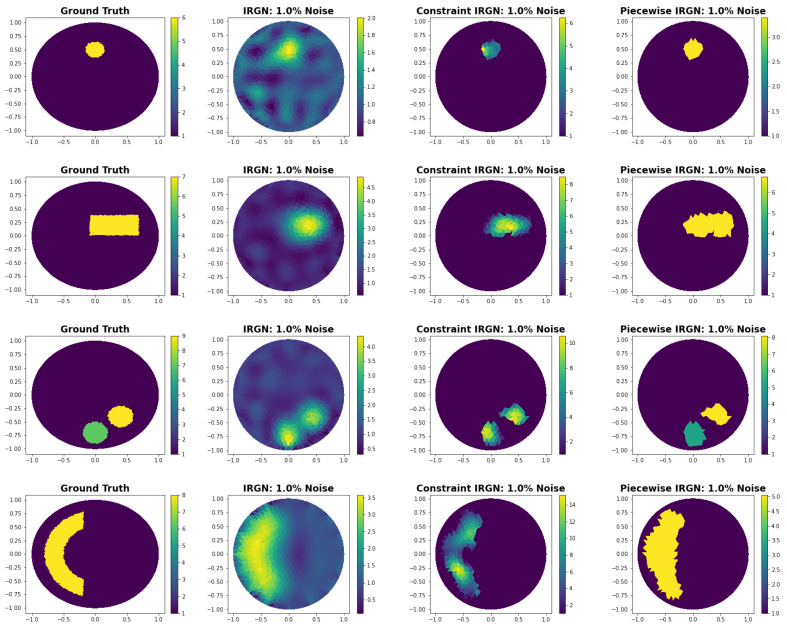
Comparison of the IRGN method with several IRGN methods that are using our shape reconstruction.

**Figure 6 entropy-25-00493-f006:**
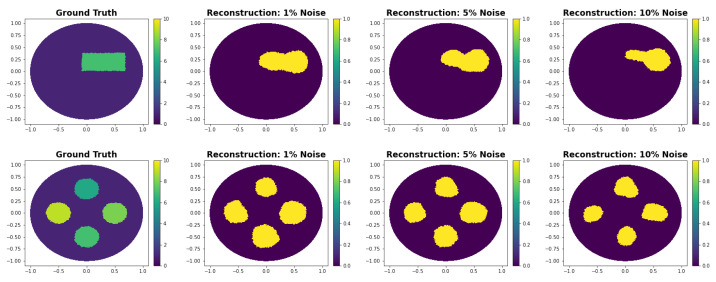
Super-resolution experiments are evaluations of the model trained on a less fine mesh. Here, the data are generated in a fine mesh and reconstructed on the same mesh. This is not an inverse crime because our method directly solves the problem in the continuous domain.

**Table 1 entropy-25-00493-t001:** Measure encoder architecture.

Network Block	Input Dimension	Output Dimension
ResNet	208	128 * scale
ResNet	128 * scale	128 * scale
ResNet	128 * scale	128 * scale
ResNet	128 * scale	64 * scale
ResNet	64 * scale	64 * scale
ResNet	64 * scale	64 * scale

**Table 2 entropy-25-00493-t002:** Point encoder architecture.

Network Block	Input Dimension	Output Dimension
ResNet	2 or 3	32 * scale
ResNet	32 * scale	64 * scale

**Table 3 entropy-25-00493-t003:** Decoder architecture.

Network Block	Input Dimension	Output Dimension
ResNet	64 * scale	32 * scale
ResNet	32 * scale	2
Softmax	2	2

**Table 4 entropy-25-00493-t004:** Noise study.

	Accuracy: Mean ± Standard Deviation			
**Noise**	**Training 0.0%**	**Training 0.1%**	**Training 1.0%**	**Training 2.0%**
Validation 0.0%	0.9737±0.0143	0.9778±0.0123	0.9623±0.0170	0.9591±0.0182
Validation 1.0%	0.9711±0.0160	0.9754±0.0139	0.9617±0.0175	0.9584±0.0187
Validation 2.0%	0.9643±0.0189	0.9686±0.0168	0.9590±0.0184	0.9566±0.0194
Validation 3.0%	0.9571±0.0219	0.9612±0.0199	0.9540±0.0199	0.9529±0.0206
Validation 4.0%	0.9503±0.0241	0.9545±0.0223	0.9485±0.0217	0.9476±0.0221
Validation 5.0%	0.9444±0.0261	0.9487±0.0247	0.9430±0.0234	0.9423±0.0236

**Table 5 entropy-25-00493-t005:** Ablation study.

	Accuracy: Mean ± Standard Deviation		
**Noise**	**Scale 1**	**Scale 2**	**Scale 4**
Validation 0.0%	0.9638±0.0203	0.9778±0.0123	0.9546±0.0274
Validation 1.0%	0.9614±0.0219	0.9754±0.0139	0.9530±0.0292
Validation 2.0%	0.9548±0.0244	0.9686±0.0168	0.9478±0.0316
Validation 3.0%	0.9477±0.0279	0.9612±0.0199	0.9423±0.0344
Validation 4.0%	0.9420±0.0307	0.9545±0.0223	0.9378±0.0371
Validation 5.0%	0.9372±0.0332	0.9487±0.0247	0.9343±0.0394

**Table 6 entropy-25-00493-t006:** Comparison with monotonicity.

	Accuracy		
**Image**	**Monotonicity: 0% Noise**	**Ours: 0% Noise**	**Ours: 3% Noise**
Figure 4, Row 1	0.9843	0.9929	0.9954
Figure 4, Row 2	0.9705	0.9871	0.9754
Figure 4, Row 3	0.9501	0.9695	0.9675
Figure 4, Row 4	0.8975	0.8910	0.8966
Figure 4, Row 5	0.7535	0.9538	0.9511

**Table 7 entropy-25-00493-t007:** Comparison with iteratively regularized Gauss–Newton method.

	$L_{2}$ Norm		
**Image**	**IRGN: 1% Noise**	**Constraint IRGN: 1% Noise**	**Piecewise IRGN: 1% Noise**
Figure 5, Row 1	1.1635	0.7413	0.7690
Figure 5, Row 2	1.8013	1.3332	1.8467
Figure 5, Row 3	2.5513	2.0178	2.0959
Figure 5, Row 4	3.0121	2.8924	3.1042
Figure 5, Row 5	3.8958	2.8328	2.9130

## Data Availability

Not applicable.

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
