# Peer review of "Implicit Solutions of the Electrical Impedance Tomography Inverse Problem in the Continuous Domain with Deep Neural Networks"

_entropy, 2023, doi:10.3390/e25030493_

Round 1
Reviewer 1 Report
1-The authors should make reference to the advantages of the proposed method over traditional methods of measurement
2-Authors should include the most recent references..
Author Response
Thank you for your time for the review and your positive feedback on our manuscript.
Concerning your comments, we made many minor changes to the manuscript, including referring to the proposed method's benefits and disadvantages and adding more references to the text.
Reviewer 2 Report
T. Strauss et al. have studied the inverse problem of Electrical Impedance Tomography, which is important for certain applications such as medical imaging. As an experimentalist working in the laboratory, I appreciate the importance of measuring the electrodes and finding the conductivity of a given object. I believe this is important work that merits being published in this journal if the mathematical rigor and numerical method has been verified by another reviewer.
Author Response
Thank you for your time for the review and your positive feedback on our manuscript.
Reviewer 3 Report
I went through the paper and it looks good. The reasonings are valid and the authors have provided enough method details to support their claims. Furthermore, the experimental sections provide enough quantitative results to further establish the correctness of the claims.
Author Response

(The authors gave the same response as above.)

Reviewer 4 Report
The manuscript presents Deep Neural Network (DNN) ML method of shapereconstruction of the conductivity ín the electrical impedance
tomography inverse problem using boundary measurements of electric
potential. The authors propose determine the shape of the object in the domain $\Omega$
with help of DNN
using separation of the unknown conductivity into two regions:the region with homogeneous conductivity and the region with non-homogeneous conductivity. They introduce the function
which classify discrete points of the domain $\Omega$ into 2 classes: homogeneous and non-homogeneous.
Then points belonging to the non-homogeneous class determine the shape of the object.
Moreover, authors combine proposed DNN method with standard gradient approach. Numerical examples show performance of DNN and gradient-based methods.
The manuscript is well written, theoretical results are
new and present interest for mathematical and ML scientific community.
I, therefore, recommend this paper for
publication, provided, however , that the authors should take into
account comments provided in the attached pdf file.

Author Response
- Please See the pdf for a proper Formate Version.
Thank you for your detailed and constructive review. We responded to each of your comments below and made the corresponding changes to our manuscript.
Page 3:, row 75: it is not clearly written that ∂Ω is the boundary of the domain Ω.
We changed the sentence to: An EIT experiment involves applying an electrical current (Neumann data) on ∂Ω, the boundary of the body Ω, to measure the corresponding electrical potential differences on ∂Ω.
Page 3:, row 114: the constant C is not defined.
We added the following sentence to the text: Here, C refers to the right hand side constant used in the following Lipschitz stability inequality ||σ1 − σ2||L∞(Ω) ≤ C||Λσ1 − Λσ2 ||⋆.
Page 5:, after row 164: when you define the forward problem you write that you have 2 types of boundary data with the same function g. This 1 is not clear: it should be then 2 different functions, g1 and g2. It seams that you di- vide your boundary ∂Ω into 2 boundaries: ∂1Ω and ∂2Ω. Then your input g1 - Neumann input current - should be on ∂1Ω, and your output data g2 - Dirichlet measured voltage - is measured at ∂2Ω. The last equation in the model problem u = 04 is not clear - why do ∂Ω you need it? Please, provide more comments on the forward problem. Where boundary electrodes are placed? At the whole boundary ∂Ω or at the part of the boundary? Where measured data is taken? We changed the injected current to g1 and measured voltage to g2 and added the following: where g1 is the injected current and g2 is the measured voltage.
Page 6, row 170: you write that you use 16 electrodes. Why do you have 208 voltage differences, not 256? Explain this.
The reason is that we have 16 electrodes. We always use two electrodes on the boundary from the current injection (source-sink). The rest of the electrodes are used for measuring the voltage differences. This is done by computing the voltage differences on two adjacent electrodes not used for current injection dur- ing the experiment. Hence, we have 13 measurements per experiment. We have 16 experiments: one for each source-sink setup of adjacent electrodes. Hence, we have 13*16 = 208 measurements. We changed the text in the following way: ”Our experiments used 16 electrodes on ∂Ω. Hence, we have 13 measure- ments per source-sink setting, which results in 208 voltage differences for all
source-sink settings that we call the measurements m.”
Page 6, row 180: can you improve your DNN algorithm to be able find small inclusions?
The section only refers to how we generated the training data set and not the actual detection rates. Here, we described a method for generating the data in such a way that it increases the robustness of the neural network, in par- ticular for detecting small inclusions. The problem was that if an inclusion in the training set is too small, then it typically gets little to none labeled point from a random point cloud. Hence, we always took the same amount of labeled random samples out of the background and inclusion. This gives the neural net- work more penalty on the detecting inclusions wrongly. The actual detection capacities of the neural network are discussed later in section 3.
Page 6, row 196: give definition of all variables in the formula for noise. Usually, noise level is given in procents. Rename level by δ.
We made the changes as proposed.
Page 7, Table 4: how accuracy was computed ? Provide formula.
We added the formula as: The accuracy for one measurement m is the ra- tio of points correctly labeled as either background or inclusion, i.e. accuracy
?
I(Predictioni=Truthi)
= i∈points , where I is the indication function.
Page 7, row 227: the training noise is 0.1%. What happens if the noise is large?
This can be seen in table 4, Noise Study. Here, we compare different training noise levels with the performance of different noise levels on a validation set. Because we found that 0.1% training noise gives the best performance for all noise levels in the validation set, we did all other experiments with a neural network trained with 0.1% training noise.
Make comment if you can get good shape reconstruction when the noise level in measured data will be larger than you are using now. We added the following text into the noise study: Note that we considered 5% as the largest noise level for the validation and test data because the reconstructions tend to become unstable, both location and shape of the inclusions, for larger noise levels.

Reviewer 5 Report
Dear Authors,
I think it is of an author's best interest to have a review with the highest amount of fair-criticism as possible, thus having his/her name associated with high-quality work. Minding the time constraints to review this paper, I spent the maximum amount of time I could on it and tried to be as critical as I could.
As a Reviewer, my general opinion is that the topic is interesting.
However, I think the manuscript is not ready yet for publication, it still needs maturing in terms of the accuracy in theory and also in formatting.
Next, see my complete review, minor and major issues are blended. The comments follow the same order as the paper.
1) I think that your Abstract is well-written. But you should be more specific regarding the methods and the implementations which are tested and proposed in your work. As I can observe, you provide more general information in abstract, please be more specific. Are the methods new in the domain or new in the research? Thank you.
2) Please provide extra information related to previous works and design methodologies. Please extend your introduction and provide more state-of the art information, for example different design methodologies and the appropriate metrics. It will increase the quality of your work and you will receive easier citations. Moreover, add a Reminder of your work. Please also be careful with abreviations!
For example:
a)A CMOS magnitude/phase measurement chip for impedance spectroscopy
b)Towards a fast and accurate eit inverse problem solver: A machine learning approach
c) An efficient point-matching method-of-moments for 2D and 3D electrical impedance tomography using radial basis functions
d)EIT reconstruction algorithms: pitfalls, challenges and recent developments
3)Please explain more Fig 1. Your explanation is a little bit weak. If I read only your text, it will not be easy to understand that you refer to this Figure. Moreover, please provide extra information in Fig 1 caption.
4) Regarding the Background, please explain more the necessity of this methodology. Please provide application example and real-world implementations. Also, the limitations of each methodology are not clearly provided.
5)Please explain more Fig. 2. We just observe schemes, what is the main achievement and what is the ResNet? It is not clear at all.
6)Please explain the reason you provide these informations in Section 2. Please provide more application specific informations. Section 2 provides generic information which can be found in many fundumental books. Please keep this information and provide extra which will be more specific in your block. In this format, it is not clear at all, it is just a combination of units and blocks. Thank you.
7)Why you choose these values for the parameters? (training dataset)
8) Have you checked your methodology over extreme cases and difficult datasets?
9) Please provide a Table in which you compare your work with previous related works. It will upgrade your work. (related metrics). Please also explain Table 2. Have you checked all these algorithms in the same dataset?
10) Please provide the difficulties of the implemented architecture. Thank you.
11) How you can use this design flow in a different application level (different types of datasets) ? Is it a simple model? Can you use it as a part of a whole general purpose system or just as a single application specific one? Please explain.
This review, is provided to the authors in order to upgrade the quality of this work. I hope to see your work published in this Journal as soon as possible.
Author Response
Thank you for your detailed and constructive review. We responded to each of your comments below and made the corresponding changes to our manuscript.
1) I think that your Abstract is well-written. But you should be more specific regarding the methods and the implementations which are tested and proposed in your work. As I can observe, you provide more general information in ab- stract, please be more specific. Are the methods new in the domain or new in the research? Thank you.
Thank you for pointing this out. We specified the novelty of the work in the abstract by adding: In this paper, we propose a piece-wise constant reconstruc- tion method that is novel in the inverse problem setting but inspired by recent work from the 3D vision community. We also extend this method into a novel constrained reconstruction method.
2) Please provide extra information related to previous works and design method- ologies. Please extend your introduction and provide more state-of the art infor- mation, for example different design methodologies and the appropriate metrics. It will increase the quality of your work and you will receive easier citations. Moreover, add a Reminder of your work. Please also be careful with abreviations!
a)A CMOS magnitude/phase measurement chip for impedance spectroscopy
Added
b)Towards a fast and accurate eit inverse problem solver: A machine learn- ing approach
Added
c) An efficient point-matching method-of-moments for 2D and 3D electrical impedance tomography using radial basis functions
Added
d)EIT reconstruction algorithms: pitfalls, challenges and recent develop- ments
Added
3)Please explain more Fig 1. Your explanation is a little bit weak. If I read only your text, it will not be easy to understand that you refer to this Figure. Moreover, please provide extra information in Fig 1 caption.
Thank you for pointing this out. We added the following to the caption: The implicit architecture is based on two main neural networks, a measurement en- coder tanking measurements as input and a point encoder that has a point in Rd as input. The outputs are added. Then a neural network called a decoder predicts the probability of a point (the input of the point encoder) being of background or anomaly type.
4) Regarding the Background, please explain more the necessity of this methodology. Please provide application example and real-world implemen- tations. Also, the limitations of each methodology are not clearly provided. We added the following to the : EIT is a great alternative to CT scans par- ticularly for neonatal brain imaging where CT can be harmful. EIT has been successfully used in medical imaging [1, 2], monitoring soil [2], or crack detection [4].
5)Please explain more Fig. 2. We just observe schemes, what is the main achievement and what is the ResNet? It is not clear at all.
We added the following addition to the text: It is among the most common building blocks in state-of-the-art networks. Here, we use the following variant: .... We also added the following text to the caption: A schematic picture of the Residual Neural Network (ResNet) versions that we use as the main building block in our implicit architecture.
6)Please explain the reason you provide these informations in Section 2. Please provide more application specific informations. Section 2 provides generic information which can be found in many fundumental books. Please keep this information and provide extra which will be more specific in your block. In this format, it is not clear at all, it is just a combination of units and blocks. Thank you.
We added a paragraph explaining why we choose the block and how they are connected: In this section, we first describe the preliminary neural network blocks. Herby, we aim to unify the basic knowledge of the terms needed to describe our network architecture. Then we describe the proposed network ar- chitecture in detail. This is followed by an explanation of how the training data is generated, what noise setting is used, and how the neural network is trained.
7)Why you choose these values for the parameters? (training dataset)
We added the following text in the corresponding section: All parameters pro- vided in this section aim to make the method reproducible. Most parameters were chosen add-hoc, but some key parameters, such as the learning rate, were iteratively improved to increase the training speed/stabiliy.
8) Have you checked your methodology over extreme cases and difficult datasets?
Yes we did. For example an image without inclusion (which works perfectly). We also tried some images that are not of the same distribution as the data that was generated for as training data set. See for example: Figure 4, row 2 or Figure 4, row 4. Both this images can not be constructed from 1-4 circular anomalies.
9) Please provide a Table in which you compare your work with previous related works. It will upgrade your work. (related metrics). Please also explain Table 2. Have you checked all these algorithms in the same dataset?
We assume that you mean Table 6 and 7. We have compared in these two ta- bles the proposed approach and existing Iteratively Regularized Gauss Newton (IRGN) approach which is a standard method in the literature. Yes these com- parisons were all computed on the same data set. Including on some extreme out of training distribution cases. This gives an advantage to the comparing algorithms as this do not require any prior training and hence the extreme cases are standard case for them.
In case you mean Table 2: It is an achitecture block of our architecture.
10) Please provide the difficulties of the implemented architecture. Thank you.
We included the following text in a new subsection: The main difficulty of the proposed method is that it assumes that the background conductivity is known. However, we note that many other methods also assume this to some extent. Further, problems with this and any other method based on neural networks are that they require an extensive training set to perform well. This is, in partic- ular, problematic when one wants to use real training data because it is costly and time intensive to create a large number of phantoms and measurements for such an approach. Hence, our training is entirely on simulated training data.
11) How you can use this design flow in a different application level (differ- ent types of datasets) ? Is it a simple model? Can you use it as a part of a
3
whole general purpose system or just as a single application specific one? Please explain.
We added the following text in the conclusions: The proposed design flow can be used for any coefficient inverse problem such as EIT. It can be easily applied to any inverse problem that aims to reconstruct unknown images or physical properties from some measurements. The only requirement for using this method is the existence of a properly defined forward model.

Round 2
Reviewer 4 Report
Authors improved their manuscript accordingly to the referee comments.
I recommend this work for publication.
Author Response
Thank you for your review.
Reviewer 5 Report
Dear Authors,
Thank you for dealing with my concerns.
I hope to see your work published as soon as possible.
Author Response
Thank you for your review.